# Unleashing the Potential of Pre-Trained Diffusion Models for Generalizable Person Re-Identification

**DOI:** 10.3390/s25020552

**Published:** 2025-01-18

**Authors:** Jiachen Li, Xiaojin Gong

**Affiliations:** College of Information Science and Electronic Engineering, Zhejiang University, Hangzhou 310027, China; lijiachen_isee@zju.edu.cn

**Keywords:** generalizable person re-identification, diffusion model, prompt learning, CLIP

## Abstract

Domain-generalizable re-identification (DG Re-ID) aims to train a model on one or more source domains and evaluate its performance on unseen target domains, a task that has attracted growing attention due to its practical relevance. While numerous methods have been proposed, most rely on discriminative or contrastive learning frameworks to learn generalizable feature representations. However, these approaches often fail to mitigate shortcut learning, leading to suboptimal performance. In this work, we propose a novel method called diffusion model-assisted representation learning with a correlation-aware conditioning scheme (DCAC) to enhance DG Re-ID. Our method integrates a discriminative and contrastive Re-ID model with a pre-trained diffusion model through a correlation-aware conditioning scheme. By incorporating ID classification probabilities generated from the Re-ID model with a set of learnable ID-wise prompts, the conditioning scheme injects dark knowledge that captures ID correlations to guide the diffusion process. Simultaneously, feedback from the diffusion model is back-propagated through the conditioning scheme to the Re-ID model, effectively improving the generalization capability of Re-ID features. Extensive experiments on both single-source and multi-source DG Re-ID tasks demonstrate that our method achieves state-of-the-art performance. Comprehensive ablation studies further validate the effectiveness of the proposed approach, providing insights into its robustness.

## 1. Introduction

Person re-identification (Re-ID) aims to match a query person’s image across different cameras based on the similarity of feature representations. Although supervised Re-ID methods based on convolutional neural network (CNN) [1,2,3,4] and visual transformer [5] have made significant advancements, their performance dramatically degenerates when applied to out-of-distribution (OOD) data that are dissimilar to training scenes. To address this problem, domain-generalizable (DG) Re-ID has garnered increasing interest in recent years. In DG Re-ID, a model is trained on one or multiple source domains and then tested on completely different and unseen domains.

Numerous DG Re-ID methods have been developed so far. Existing studies concentrate on domain-invariant and domain-specific feature disentanglement [6,7,8], normalization and domain alignment [9,10,11,12,13,14,15], or employ meta-learning [10,16,17,18] and other techniques like semantic expansion [19,20] and sample generation [21] to enhance the generalization capability of Re-ID models. Although various techniques have been designed, almost all methods learn feature representations within discriminative or contrastive learning frameworks, which are considered unable to prevent shortcut learning [22], leading to suboptimal performance.

Recently, diffusion models, such as Imagen [23] and Stable Diffusion [24], have demonstrated remarkable capabilities in image synthesis and other generative tasks. Moreover, their potential for representation learning has been increasingly recognized in recent studies [25,26,27]. On the one hand, pre-training on extensive multi-modal data equips diffusion models with rich semantic information, enabling exceptional generalization and robustness in out-of-distribution scenarios [28]. On the other hand, the denoising process inherently promotes the learning of meaningful semantic representations [29]. Inspired by these observations, this work explores leveraging a diffusion model to enhance the generalization ability of representations initially learned within discriminative and contrastive learning frameworks, thereby improving domain-generalizable person Re-ID.

To this end, we propose a generalizable Re-ID framework comprising a baseline discriminative and contrastive Re-ID model, a generative diffusion model, and a conditioning scheme that bridges the two models. Unlike existing diffusion-based representation learning techniques, which either jointly train a denoising decoder and a classifier with a shared encoder [26,30] or intertwine feature extraction and feature denoising [31], our approach opts to preserve the integrity of the pre-trained diffusion model, as shown in Figure 1a–d. This choice allows the semantic knowledge embedded in the diffusion model to be effectively transferred to the Re-ID baseline. Furthermore, unlike SODA [27] and DIVA [32], which adopt an instance-level conditioning mechanism, we introduce a new conditioning scheme that is ID-wise and explicitly aware of ID correlations.

More specifically, as recognized in knowledge distillation [33] and illustrated in Figure 1e, dark knowledge embedded in the logits of classifiers captures the relationships among IDs, reflecting nuanced similarities and differences that go beyond hard class labels. Therefore, we design a correlation-aware conditioning scheme that integrates classification probabilities with learnable ID-wise prompts as the guidance of the diffusion model. This conditioning scheme makes the diffusion model less sensitive to intra-ID variances and background interference compared to instance-level conditioning and more expressive and adaptable than one-hot ID labels, enabling it to better capture complex inter-ID relationships and improve generalizable Re-ID performance. Additionally, we employ LoRA [34] adapters to enable the parameter-efficient fine-tuning (PEFT) of the diffusion model alongside the full fine-tuning of the Re-ID model. This approach allows the diffusion model to adapt effectively and efficiently to Re-ID data while preserving the knowledge embedded in the pre-trained diffusion model, thereby mitigating the risk of catastrophic forgetting often associated with full model fine-tuning [35].

The main contributions of this work are summarized as follows:We investigate the feasibility of leveraging a pre-trained diffusion model as an expert to enhance generalizable feature learning for DG Re-ID by collaboratively training a discriminative Re-ID model and efficiently fine-tuning a generative diffusion model;We propose a simple yet effective correlation-aware conditioning scheme that combines the dark knowledge embedded in ID classification probabilities with learnable ID-wise prompts to guide the diffusion model, unleashing its generalization knowledge to the discriminative Re-ID model through gradient feedback;Extensive experiments on both single-source and multi-source DG Re-ID tasks demonstrate the effectiveness of our approach, achieving state-of-the-art performance. Additionally, plenty of ablation studies are conducted to provide a comprehensive analysis of the proposed method.

## 2. Related Work

### 2.1. Generative Diffusion Models

Diffusion models [36,37], which simulate a Markov chain to learn the transition from noise to a real data distribution, have shown remarkable performance in generation tasks. Representative diffusion models include Imagen [23], stable diffusion [24], and DiT [38]. Imagen predicts noise in a pixel space and generates high-resolution outputs using super-resolution modules. In contrast, stable diffusion and DiT denoise images in latent spaces, significantly reducing computation costs. Specifically, stable diffusion maps an image into the latent space via a pre-trained variational autoencoder (VAE) and predicts noise with a U-Net structure [39] containing cross-attention modules to fuse conditions. DiT further replaces the U-Net with visual transformers and improves the condition injection with the adaLN-zero strategy for scalable high-quality image generation. Considering computational efficiency, we choose to adopt stable diffusion in this work.

### 2.2. Diffusion Models for Representation Learning

Although diffusion models are primarily designed for generation tasks, their ability to learn semantic representations has also been recognized in recent years [29]. For example, Baranchuk et al. [25] and DDAE [40] leverage the intermediate activations of pre-trained diffusion models as features for segmentation and classification, respectively. HybViT [26] and JDM [30] jointly learn discriminative and generative tasks with a shared encoder to enhance feature representation. SODA [27] turns diffusion models into strong self-supervised representation learners by imposing a bottleneck between an encoder and a denoising decoder. DIVA [32] employs the feedback of a frozen pre-trained diffusion model to boost the fine-grained perception capability of CLIP [41] via a post-training approach. Additionally, diffusion models are exploited as zero-shot classifiers [42,43] by estimating noise given the class names, such as conditions, exhibiting great generalization robustness in out-of-distribution scenarios [28]. Inspired by these studies, we explore the utilization of a pre-trained diffusion model to enhance representation learning for the generalizable Re-ID tasks.

### 2.3. Diffusion Models for Person Re-ID

Diffusion models have also been applied to various person Re-ID tasks. For instance, VI-Diff [44] employs a diffusion model to enhance visible-infrared Re-ID by generating new samples across modalities, thereby reducing the annotation cost of paired images. Diverse person [45] proposes a diffusion-based framework to edit original dataset images with attribute texts, efficiently generating high-quality text-based person search datasets. PIDM [46] also focuses on new data generation, using body pose and image style as guidance. Asperti et al. [47] decouple the person ID from other factors like poses and backgrounds to control new image sample generation. These works share a common characteristic of modifying existing data or generating new data for Re-ID related tasks. Additionally, DenoiseReID [31] unifies feature extraction and feature denoising to improve feature discriminative capabilities for Re-ID. PISL [48] proposes a spatial diffusion model to refine patch sampling to enhance unsupervised Re-ID. PSDiff [49] formulates the person search as a dual denoising process from noisy boxes and Re-ID embeddings to ground truths. In contrast to these, we focus on generalizable representation learning assisted by the feedback back-propagated from a pre-trained diffusion model.

### 2.4. Generalizable Person Re-ID

Generalizable person Re-ID has been extensively studied over the past years. Existing methods can be roughly categorized into the following groups: domain-invariant and specific feature disentanglement [6,7,8], normalization and domain alignment [9,10,11,12,13,14,15], learning domain-adaptive mixture-of-experts [13,50,51], meta-learning [10,16,17,18], semantic expansion [19,20], large-scale pre-training [52,53,54], and so on. While various mechanisms have been designed, most of these methods learn feature representations within discriminative [6,9] or contrastive learning [16,52] frameworks. In contrast, we aim to leverage a pre-trained generative diffusion model to enhance the domain-invariant feature learning for more robust generalizable Re-ID.

## 3. Diffusion Preliminaries

In this section, we briefly recap the preliminaries of classical diffusion models [36,37]. The diffusion models are generative models defined on a Markov chain, where the forward and reversed processes are modeled using a forward diffusion kernel (FDK) q(xt|xt−1) and a learnable reverse diffusion kernel (RDK) pθ(xt−1|xt). In the forward process, a real sample x0 is gradually disturbed towards a final state xT that is quite close to a pure Gaussian noise via a FDK. In the reverse process the RDK is trained to denoise from xT to x0. The real distribution of x0 can be constructed using the integral over each possible path dx1:T with an optional condition c as guidance:(1)pθ(x0|c)=∫x1:Tp(xT)∏t=1Tpθ(xt−1|xt,c)dx1:T.

To estimate the denoising model parameter θ, the negative log-likelihood loss −logpθ(x0|c) should be minimized, but the integral is infeasible. Thus, the variational lower bound LELBO is optimized as an alternative:(2)LELBO=Eqq(x1:T|x0)pθ(x0:T,c)≥−logpθ(x0|c),
where LELBO can be further expanded and simplified to show its essence [36,37], which actually learns to predict the added noise at each timestep *t* by the mean square error:(3)Lmse=Et||ϵt−ϵθ(xt,c)||2,
where xt is the noisy input, which follows the following equation:(4)xt=α¯tx0+1−α¯tϵt.
Noise ϵt is sampled from the isotropic Gaussian distribution N(0,I). *t* is sampled from a series of timesteps {1,2,⋯,T}, which controls the strength of noise by selecting scheduled diffusion rate α¯t.

## 4. The Proposed Method

As illustrated in Figure 2, the overall framework comprises a baseline Re-ID model, a pre-trained diffusion model, and a correlation-aware conditioning scheme that bridges the two models. The Re-ID model learns feature representations by optimizing a discriminative ID loss and a prototypical contrastive loss. The ID classification probabilities generated from the Re-ID model are used to inject the dark knowledge [33] of different IDs into a correlation-aware condition that guides the diffusion process. Simultaneously, the gradients of the diffusion model are back-propagated through the condition to the Re-ID model, transferring generalization knowledge to enhance Re-ID feature learning. During test time, only the image encoder of the Re-ID model is used for feature extraction.

### 4.1. The Baseline Re-ID Model

The baseline Re-ID model comprises an image encoder Eψ, together with a classifier supervised by discriminative ID loss Lid and additional prototypical contrastive loss (PCL) Lpcl. Our image encoder is constructed based on the pre-trained CLIP [41] image encoder, appended with a batch normalization neck (BNNeck) [1]. Leveraging the CLIP encoder enables our model to acquire a certain level of generalization abilities, attributed to its extensive language-image pre-training.

Formally, given an input image x and its feature z=Eψ(x)∈R1×d, we define Lid and Lpcl as follows:(5)Lid=−∑j=1Nqjlogexp(zwj⊤)∑k=1Nexp(zwk⊤),(6)Lpcl=−logexp(zM[y]⊤/τ)∑k=1Nexp(zM[k]⊤/τ),
where *N* is the number of IDs, wj∈R1×d denotes the weights of the *j*-th ID in the classifier, qj denotes the smoothed ground-truth label, and τ is a temperature factor. Moreover, M∈RN×d represents the prototypical memory bank. Each entry in M is initialized with the feature centroid of images belonging to the corresponding ID at the beginning of every epoch. Subsequently, it is updated in a moving average manner with momentum γ:(7)M[y]←γM[y]+(1−γ)zhard,
in which *y* denotes the ID label of the image x, and zhard is the hardest sample [55] of the corresponding ID within a batch.

Then, the baseline Re-ID model learns feature representations by optimizing the following loss:(8)LReID=Lid+Lpcl.

### 4.2. The Generative Diffusion Model

Our work aims to leverage both the semantic knowledge acquired from a pre-trained diffusion model and the assistance provided by the denoising process to enhance the feature learning capabilities of the baseline Re-ID model’s encoder. To this end, rather than directly utilizing intermediate activations of the diffusion model as features [25,40] or training a denoising decoder alongside the Re-ID model’s classifier using a shared encoder like [26,30], we opt to employ a complete pre-trained diffusion model and adapt it to Re-ID data using LoRA [34].

More specifically, we adopt the pre-trained stable diffusion model [24] in our work. This diffusion model employs a variational autoencoder (VAE) composed of an encoder Evae and a decoder Dvae to map an input image into a latent space, facilitating a more efficient diffusion process. Moreover, it integrates cross-attention layers into a U-Net [39] architecture Eθ to denoise latent features, enabling the incorporation of various types of conditions. To effectively adapt the diffusion model to Re-ID data while preserving the generalization capabilities acquired during pre-training, we employ LoRA [34] adapters for fine-tuning.

LoRA [34] adapters are only applied to the transformation matrices in the attention layers, including the query, key, value and output transformation matrices in attention computation, and the linear transformation matrices in feed-forward networks, as shown in Figure 2. Formally, the LoRA [34] adapters introduce low-rank projection matrices A∈Rdin×r and B∈Rr×dout to create modifications on original output features as follows:(9)h′=hW+1rhAB,
where h∈R1×din and h′∈R1×dout denote the input and output features, respectively. W∈Rdin×dout denotes each possible original transformation matrices mentioned before. *r* is called rank, which controls the size of the low-dimension space. din and dout are the dimensions of input and output features, respectively, where r≪min(din,dout). Throughout the entire training process, we freeze the diffusion model while keeping the A and B matrices of LoRA [34] adapters trainable. The purposes of utilizing LoRA [34] adapters in the diffusion model are two-fold: (1) it reduces computational overhead compared with other fine-tuning methods and (2) mitigates the risk of the catastrophic forgetting [35] of learned knowledge from pre-training. We will further discuss the effectiveness of LoRA [34] adapters in Section 5.4.

When an image x is input to the image encoder of the Re-ID model, the image is also resized to an image x0 to match the input size of the diffusion model. Then, x0 is fed into the VAE encoder Evae to produce a latent feature F0=Evae(x0). With a random noise ϵt sampled from the isotropic Gaussian distribution N(0,I), the noisy feature Ft at the timestep *t* is obtained, as mentioned in Equation (Equation 4):(10)Ft=α¯tF0+1−α¯tϵt.

Afterward, Ft and its corresponding condition c (to be introduced in the following subsection) are forwarded to U-Net Eθ to minimize the noise estimation error expectation, as mentioned in Equation (Equation 3) with new notation Ldif:(11)Ldif=Et||ϵt−Eθ(Ft,c)||2.

### 4.3. The Correlation-Aware Conditioning Scheme

We design a conditioning scheme to bridge the Re-ID model and the diffusion model, enabling mutual interaction. This scheme enables the use of information from the Re-ID model to guide the diffusion process while simultaneously enabling the feedback from the diffusion model to be back-propagated to improve the Re-ID model. A straightforward conditioning scheme is to take the instance feature encoded by the Re-ID model as the condition, similarly to SODA [27] and DIVA [32]. However, such instance-level features are sensitive to intra-ID variations and background changes, making them less robust to domain shifts and resulting in suboptimal generalization performance.

Therefore, we opt to design the condition in an ID-wise manner. Notably, the ID classification probabilities produced by the baseline Re-ID model not only indicate the ID class to which an image belongs but also encapsulate dark knowledge about the correlations among different IDs, which has been shown to enhance generalization capabilities [56,57,58]. Building on this insight, we incorporate the classification probabilities along with a set of learnable ID prompts to define the condition.

Specifically, we create a set of learnable ID prompts Eid=e1,...,ej,...,eN∈RN×d. The prompt ej∈R1×d corresponds to the *j*-the ID, where *d* is the dimension of each prompt. Then, the condition c∈R1×d for the image instance x is generated by a linear combination of all ID prompts weighted by the classification probability of the input instance: (12)c=∑j=1Npjej=∑j=1Nexp(zwj⊤/τc)∑k=1Nexp(zwk⊤/τc)ej,
where pj denotes the probability of the instance, with x being classified into the *j*-th ID class, as defined on the right side of the equation. z and wj are the image feature and the *j*-th classifier, respectively, as defined in Section 4.1. τc is a temperature factor that regulates the probability distribution. Unlike the linear combination used in MVI^2^P [59], which aims to integrate multiple discriminative feature maps for Re-ID training, our approach generates conditions that guide the diffusion model.

We refer to this design as the correlation-aware conditioning scheme. The correlation-aware condition c aims to describe current image instance x with all possible IDs. It improves the robustness of representation learning via combining the ID-wise prompts with the ID classification probabilities. Despite the simplicity of this design, our experiments demonstrate that it outperforms more intricate alternative designs.

### 4.4. The Entire Training Loss

Standing on a comprehensive view, the total loss Ltotal in our framework is a composite of the Re-ID loss LReID and the diffusion loss Ldif, which is formulated as follows:(13)Ltotal=LReID+λLdif,
where λ is a balancing factor.

To this end, we have introduced all related losses in both discriminative and generative objectives. Re-ID loss LReID integrates ID loss Lid in Equation (Equation 5) and prototypical contrastive loss Lpcl in Equation (Equation 6). These losses are designed to optimize the discriminative capabilities of the Re-ID model. Diffusion loss Ldif in Equation (Equation 11) focuses on minimizing the noise estimation error expectation in the latent space during the diffusion process. This loss ensures that the diffusion model is able to effectively contribute to the learning of generalizable features guided by our proposed correlation-aware conditioning scheme. By combining LReID and Ldif, Ltotal improves discriminative learning with generative capabilities, enhancing the Re-ID model’s performances across diverse domains.

## 5. Experiments

### 5.1. Datasets and Evaluation Protocols

We conduct experiments on the following datasets: Market1501 [60], DukeMTMC-reID [61], MSMT17 [62], and CUHK03-NP [63], abbreviated as MA, D, MS, and C3, respectively. Table 1 presents detailed information of each dataset.

The performance of generalizable Re-ID is evaluated using both single-source and multi-source generalization protocols. In the single-source protocol, the Re-ID model is trained on one dataset and tested on another target dataset. For example, we denote the experiment as MS→MA when training on MSMT17 [62] and testing on Market1501 [60], and likewise for others. In the multi-source protocol, a leave-one-out strategy is employed, where one dataset is used for testing while the training sets of remaining datasets are used for training. In both protocols, we adopt the mean average precision (mAP) and cumulative matching characteristic (CMC) at Rank-1 (R1) as evaluation metrics without applying re-ranking post-processing [63].

### 5.2. Implementation Details

We implement our model in PyTorch 1.13.1 [64] and conduct experiments on an NVIDIA RTX A6000 GPU. We adopt the image encoder of the pre-trained CLIP ViT-B-16 [41] for our baseline Re-ID model, in which the patch projection layer is frozen for stability while the other parameters are trainable. The input image size for the Re-ID model is 256×128, and the dimension of the encoded feature is 512. The momentum γ utilized for updating the prototypical memory bank is set to 0.2, and the temperature factor τ in the PCL loss is set to 0.01. For the diffusion model, we adopt the pre-trained weight stable-diffusion-v1-5 [24] on Huggingface. The input image size for diffusion is 128×128. The rank *r* of the LoRA adapters is set to 8 for all single-source experiments except for those trained on MSMT17 [62]. For single-source experiments trained on MSMT17 [62] and all multi-source experiments, *r* is set to 32. In the correlation-aware conditioning scheme, the prompt dimension is set to 768 to match the diffusion model. The temperature factor τc is closely related to the size of the training datasets. Accordingly, it is set to 0.1 for training on CUHK03-NP [63], 0.6 for training on Market1501 [60] and DukeMTMC-reID [61], and 1.0 for training on MSMT17 [62] and multi-source settings. Moreover, balancing factor λ in the entire loss is set to 1.

During training, random horizontal flipping, cropping, and erasing [65] augmentations are used for the input of the Re-ID model, while no data augmentation is used for the diffusion model. We train our entire framework for 60 epochs using the Adam optimizer [66] with a base learning rate of 5×10−6 regulated by a step scheduler, which starts at a learning rate of 5×10−7 with a linear warmup in the first 10 epochs. The learning rate is multiplied with a factor of 0.1 at the 30th and 50th epoch. We adopt a weight decay of 1×10−4. The batch size is set to 64, employing a PK-sampling strategy [67] with randomly selected 16 IDs and 4 samples per ID.

### 5.3. Comparison with State-of-the-Arts

#### 5.3.1. Single-Source DG Re-ID

We first compare the proposed method, named DCAC, with several representative and state-of-the-art methods using the single-source protocol. We conduct all possible single-source generalization experiments with four datasets for comprehensive evaluations. The results are separately presented in Table 2 and Table 3. The source domain is selected from Market1501 [60] and DukeMTMC-reID [61] in Table 2 and from MSMT17 [62] and CUHK03-NP [63] in Table 3. The results show that some previous methods, such as TransMatcher [68] and MDA [17], exhibit strong performance in MS→MA generalization. However, their performance significantly deteriorates when applied to the more challenging MA→MS generalization task. In contrast, our approach demonstrates more balanced improvements in all single-source generalization tests without requiring a larger input size as in TransMatcher [68] or test-time updating as in MDA [17].

#### 5.3.2. Multi-Source DG Re-ID

More experiments are carried out using the multi-source protocol to further validate the effectiveness of the proposed approach. The results are presented in Table 4. Although some latest works like UDSX [20] and SALDG [51] demonstrate better performances when MA is selected as the target domain, their improvements are limited when generalizing to other target domains, where the average performances are 43.4% on mAP and 61.7% on Rank-1 for UDSX [20] and 40.0% on mAP and 58.6% on Rank-1 for SALDG [51], respectively. In contrast, our approach presents more balanced enhancements across all four target domains, with an average mAP of 46.4% and Rank-1 of 63.9%, surpassing the performance of UDSX [20] by a fair margin of 3.0% on mAP and 2.2% on Rank-1. In particular, with respect to the most challengeable dataset MS, our approach greatly outperforms the current best performance by 5.9% on mAP and 9.1% on Rank-1. These results show the effectiveness of our approach on multi-source DG Re-ID with state-of-the-art performances.

### 5.4. Ablation Studies

#### 5.4.1. Effectiveness of the CLIP-Based Re-ID Model

Our baseline Re-ID model is built upon the pre-trained CLIP image encoder and fine-tuned on Re-ID datasets using both a discriminative ID loss and a prototypical contrastive loss. Benefiting from pre-training on extensive text-image paired data, the CLIP encoder equips our baseline model with a certain level of generalization capability, as demonstrated in Table 2 and Table 3.

#### 5.4.2. Effectiveness of the Diffusion Model Assistance

We conduct a series of experiments to validate the effectiveness of the diffusion model for learning generalizable representations. To investigate whether the knowledge learned from pre-training or the denoising process itself is beneficial, we carry out a comparison among the following model variants: (1) using the baseline Re-ID model without diffusion, (2) using the pre-trained diffusion model while keeping it frozen, (3) using the pre-trained diffusion model with LoRA for adaptation, (4) using the pre-trained diffusion model with only the output blocks trainable, (5) using the pre-trained diffusion model with only the middle and output blocks trainable, (6) using the pre-traind diffusion model with all parameters trainable, and (7) using a randomly initialized diffusion model with all parameters trained from scratch. The results are presented in Table 5.

According to the results, we observe that fully freezing the diffusion model prevents it from effectively enhancing generalization abilities and may even slightly harm it. We attribute this to the domain gap between the diffusion model’s pre-training dataset and the Re-ID dataset. Since the diffusion model lacks specific knowledge about Re-ID, it provides invalid feedback.

A classical solution for adapting downstream knowledge is to freeze shallow blocks but train the deep blocks of the model, which is denoted as partial fine-tuning. The diffusion U-Net contains input, middle, and output blocks. We gradually unfreeze each block of the U-Net, denoted as partial fine-tuning 1 and 2, where in 1 the output blocks are trainable, and in 2, both the middle and output blocks are trainable. According to the results, partial fine-tuning presents a certain level of improvement on generalization and achieves the best on the MS→MA Rank-1 metric, with the middle and output blocks trainable. But it fails to maintain its advantage on more challengeable MA→MS generalization, in which the source domain is limited with fewer IDs and samples, and it is more likely to be overfitted. This reveals that partial fine-tuning is unable to effectively preserve pre-trained generalized knowledge during downstream adaptation.

When the pre-trained diffusion model is fine-tuned with all trainable parameters (from all input, middle, and output blocks) or when a randomly initialized diffusion model is trained without pre-training, both variant models adapt sufficiently to the Re-ID data, resulting in better performance compared to a fully frozen model. However, these models either suffer from the significant forgetting of pre-trained knowledge [35] or lack any pre-training knowledge, leading to only limited improvement.

In contrast, the approach utilizing LoRA, which only fine-tunes the low-rank adapters of the attention layers of the pre-trained diffusion model on Re-ID data, achieves balanced and significant enhancement in generalization ability across different target domains. This result highlights that the synergy between pre-trained knowledge and the diffusion process contributes most effectively to improving generalization, and illustrates that the LoRA-based fine-tuning best fits our framework.

#### 5.4.3. Ablations on Computational Overhead

Table 6 studies the computational overhead of different variants of the diffusion model’s fine-tuning, including the major approaches mentioned in Section 5.4.2 with a batch size of 64. In the training stage, the baseline model presents the optimal efficiency with 140.3 ms latency and 1.46 TFLOPs in forward propagation and 7.58 GB memory consumption with 85.94 M trainable parameters, but it suffers from limited generalization performance. Frozen fine-tuning even fails to surpass the baseline on either generalization performance or computational efficiency, with a slight increase in trainable parameters due to the incorperation of learnable ID prompts. In addition, other fine-tuning methods like partial and full fine-tuning do not demonstrate effective enhancements on generalization, although more parameters are allowed to be optimized. Note that the TFLOPs remain unchanged as 8.50 for frozen, full, and partial fine-tuning, since the count of floating point operations in forward propagation is not interfered by parameter freezing or not.

Differently, our LoRA-based strategy achieves a great tradeoff between the computational overhead and the generalization performance, enabling it to be be trained with acceptable cost growth due to newly introduced adapters for the best performance. In the inference stage, only the Re-ID image encoder Eψ with the updated parameters is required to extract person features; thus, the overhead of the diffusion model is dropped. Moreover, the cost can be further reduced without gradient computation and the classifiers in training.

#### 5.4.4. Effectiveness of the Conditioning Scheme

In our design, we claim that the proposed correlation-aware conditioning scheme is the most appropriate mechanism to guide generalization feedback from the pre-trained diffusion model, where each condition is generated by linear combination of multiple learnable ID-wise prompts weighted by the classification probabilities.

In Table 7, we compare different conditioning schemes. It is obvious that the instance-wise condition only provides a tiny contribution to generalization improvements. To further validate that the correlation among IDs, i.e., dark knowledge, is the key for transfering generalization knowledge from the diffusion model, we conduct an experiment on a simplified class-wise condition, where softmax weighting in Equation (Equation 12) is replaced by one-hot selection, which only keeps the probability score of the corresponding class and resets others to zero. The dark knowledge that exists in probability distributions is therefore erased. From the results, we find that the class-wise condition that only considers a single ID cannot effectively enhance the generalization capability, which even deteriorates the baseline on MS→MA generalization, whereas our correlation-aware condition brings the most salient enhancement, validating the importance of dark knowledge in condition generation.

Moreover, we conduct more experiments on the test sets of source domains, aiming to further investigate the performance on source domain data distribution. As demonstrated in Table 8, our correlation-aware conditioning scheme does not present obvious performance degradation on source domains, meaning that our approach indeed refines the capability of generalization and also preserves the performance on source domains.

#### 5.4.5. Impact of the Hyper-Parameters

Table 9 investigates the impact of rank *r* in the LoRA adapters. For simplicity, we choose MA as the representative of small-scale source domains, i.e., MA, D, and C3, to analyze the rank value on MA→MS generalization. We observe that a lower rank r=8 is optimal for training on small datasets. For the dataset at larger scales, i.e., MS, we tested it with respect to MS→MA generalization and found that the higher rank r=32 was optimal.

#### 5.4.6. Impact of More Intricate Conditioning Schemes

We investigate more intricate alternative designs of the conditioning scheme by employing further transformations on the basis of the original correlation-aware conditioning through linear combination. These new methods focus on the influences of the other two operations that frequently appear in neural networks, that is, non-linear mapping and normalization, instead of the linear operation only.

In Table 10, we compare these alternatives with the baseline and standard DCAC. “Non-linearity” denotes that we apply the SiLU [77] activation function, which is the same as the one activating latent features in the diffusion model, on the correlation-aware conditions to introduce non-linearity. “BatchNorm” denotes that a batch normalization layer [78] is appended after the linear combination of the ID-wise prompts to eliminate the internal covariate shift. “ConditionNet” denotes that a multi-layer perceptron network, including the mixture of linear, non-linear, and normalization layers, is employed on generated conditions. Surprisingly, we find that the more complicated variants do not seem to provide further improvements in generalization capability compared with the simple but effective correlation-aware conditioning.

#### 5.4.7. Visualization Results

In Figure 3, we use GradCAM [79] visualization to investigate the Re-ID model’s capability of capturing correlations across different IDs, which reflects the generalization capability of the Re-ID model. Specifically, we select several visually similar IDs #27, #76, and # 649 and compute the activation maps under another similar ID #28. A well-generalized Re-ID model is expected to focus on similar ID-relevant areas even if the ID class and the image is not consistent.

From the results, we observe that the baseline model mainly focuses on background areas but the person bodies are almost ignored, indicating its poor ability to learn ID correlations. When adopting the diffusion knowledge feedback, the attentive areas are rectified to shared ID-relevant attributes such as the purple T-shirt and black trousers. Furthermore, using our correlation-aware conditioning scheme helps covering more body parts and reducing perception on background areas, which shows the effectiveness of our approach.

## 6. Conclusions

In this work, we explore the feasibility of leveraging a pre-trained diffusion model to enhance generalizable feature learning for DG Re-ID. By adopting a simple yet effective correlation-aware conditioning scheme, we utilize the ID classification probabilities to guide the diffusion model for generalization knowledge unleashing and transferring towards the Re-ID model via gradient feedback. Through extensive experimentation on both single- and multi-source DG Re-ID settings, our approach demonstrates its effectiveness by achieving state-of-the-art performance levels.

## Figures and Tables

**Figure 1 sensors-25-00552-f001:**
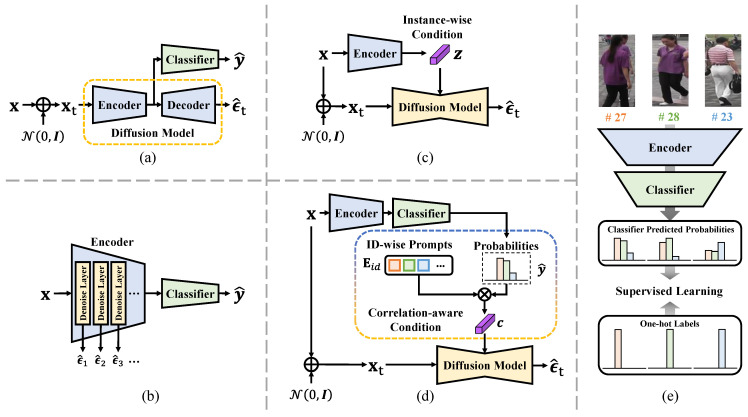
Illustration of different diffusion-based representation learning designs. (**a**) A separate denoising decoder and a classifier with a shared encoder, (**b**) intertwined feature extraction and feature denoising, (**c**) a diffusion model with a separate image encoder for instance-wise conditioning, and (**d**) a diffusion model and a classification model bridged by a correlation-aware ID-wise conditioning scheme. In addition, (**e**) illustrates that the dark knowledge embedded in the logits of classifiers is able to capture the ID relationships, including nuanced similarities and differences beyond the hard ID labels, which helps generate better conditions to guide the diffusion model.

**Figure 2 sensors-25-00552-f002:**
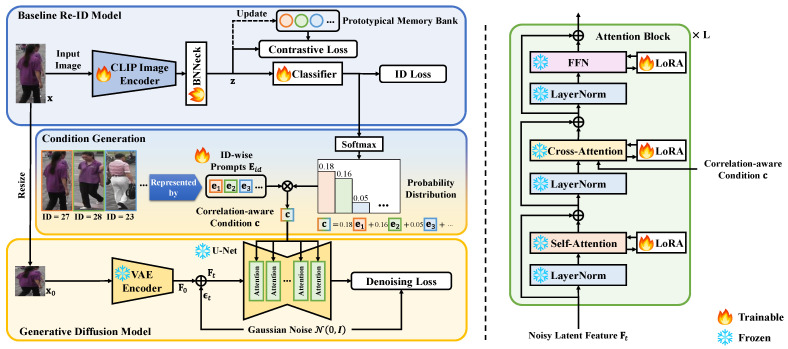
An overview of the proposed framework. It consists of a baseline Re-ID model, a pre-trained diffusion model, and a correlation-aware conditioning scheme based on learnable ID-wise prompts. The Re-ID model is built upon the pre-trained CLIP image encoder [41] and a BN Neck [1], optimized by an ID loss and a prototypical contrastive loss. The diffusion model is constructed on via pre-trained stable diffusion [24], with LoRA [34] for efficient adaptation. The informative classification probabilities predicted by the Re-ID model is employed to produce a correlation-aware condition to guide the diffusion model for unleashing specific knowledge of generalization, with gradients back-propagated to the Re-ID model for enhanced generalizable feature learning.

**Figure 3 sensors-25-00552-f003:**
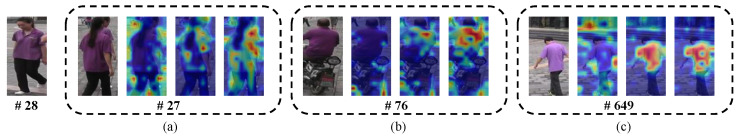
GradCAM [79] visualization of several visually similar IDs selected from the Market1501 [60] dataset. In groups (**a**–**c**), the activation maps are computed with the images of ID #27, #76, and #649 under ID #28, respectively, which reflects the Re-ID model’s capability of capturing correlations among IDs. From left to right, each group contains the original image and the activation maps of the baseline model, the instance-wise condition-guided model, and our correlation-aware-condition-guided model, respectively.

**Table 1 sensors-25-00552-t001:** The statistics of four public Re-ID datasets and their composition on different splits, including the training set (denoted by “Train”) and testing set (denoted by “Query” and “Gallery”).

Dataset	Cameras	IDs	Train	Query	Gallery
Market1501 [60]	6	1501	12,936	3368	15,913
DukeMTMC-reID [61]	8	1812	16,522	2228	17,661
MSMT17 [62]	15	4101	32,621	11,659	82,161
CUHK03-NP [63]	2	1467	7365	1400	5332

**Table 2 sensors-25-00552-t002:** Comparison with SOTA methods on the single-source DG Re-ID setting with source domains Market1501 [60] and DukeMTMC-reID [61]. † indicates that the method uses target domain data for test-time updating. ‡ indicates that the method uses input image sizes larger than 256 × 128. The best and second-best results are marked with bold and underline, respectively. Data are collected from corresponding works.

Model	MA→D	MA→MS	MA→C3	D→MA	D→MS	D→C3
mAP	R1	mAP	R1	mAP	R1	mAP	R1	mAP	R1	mAP	R1
SNR [6]	33.6	55.1	-	-	-	-	33.9	66.7	-	-	-	-
CBN [9]	-	-	9.5	25.3	-	-	-	-	9.5	25.3	-	-
QAConv [69]	28.7	48.8	7.0	22.6	-	-	27.2	58.6	8.9	29.0	-	-
TransMatcher ‡ [68]	-	-	18.4	47.3	-	-	-	-	-	-	-	-
MetaBIN [10]	33.1	55.2	-	-	-	-	35.9	69.2	-	-	-	-
DTIN-Net [11]	36.1	57.0	-	-	-	-	37.4	69.8	-	-	-	-
QAConv-GS [70]	-	-	15.0	41.2	-	-	-	-	-	-	-	-
MDA † [17]	34.4	56.7	11.8	33.5	-	-	38.0	70.3	-	-	-	-
Li et al. [71]	-	-	21.8	47.5	-	-	-	-	-	-	-	-
SuA-SpML [18]	34.8	55.5	11.1	30.1	-	-	36.3	65.8	13.6	37.8	-	-
DIR-ReID [7]	33.0	54.5	-	-	-	-	35.2	68.2	-	-	-	-
GN [14]	34.0	52.3	10.3	28.6	14.5	14.4	34.3	64.3	12.3	33.8	10.3	10.2
GN+SNR [14]	34.7	55.4	-	-	15.2	15.1	36.9	68.5	-	-	11.5	11.0
PAT [72]	-	-	18.2	42.8	-	-	-	-	-	-	-	-
LDU [15]	38.0	59.5	13.5	35.7	18.2	18.5	**42.3**	**73.2**	16.7	44.2	14.2	14.2
MTI [73]	36.4	57.8	-	-	16.2	16.3	38.2	70.5	-	-	13.3	13.3
Baseline (ours)	47.8	67.3	22.0	50.1	30.0	30.4	41.7	70.5	19.0	46.9	22.1	23.1
DCAC (ours)	**49.5**	**69.1**	**23.4**	**52.1**	**32.5**	**33.2**	**42.3**	71.5	**19.7**	**47.4**	**23.0**	**23.5**

**Table 3 sensors-25-00552-t003:** Comparison with SOTA methods on the single-source DG Re-ID setting, where the source domains are MSMT17 [62] and CUHK03-NP [63]. † indicates that the method uses target domain data for test-time updating. ‡ indicates that the method uses input image size larger than 256 × 128. The best and second-best results are marked with bold and underline, respectively. Data are collected from the corresponding works.

Model	MS→MA	MS→D	MS→C3	C3→MA	C3→D	C3→MS
mAP	R1	mAP	R1	mAP	R1	mAP	R1	mAP	R1	mAP	R1
PCB [74]	26.7	52.7	-	-	-	-	-	-	-	-	-	-
MGN [75]	25.1	48.7	-	-	-	-	-	-	-	-	-	-
OSNet-IBN [76]	37.2	66.5	45.6	67.4	-	-	-	-	-	-	-	-
SNR [6]	41.4	70.1	50.0	69.2	-	-	-	-	-	-	-	-
CBN [9]	45.0	73.7	-	-	-	-	-	-	-	-	-	-
TransMatcher ‡ [68]	52.0	**80.1**	-	-	-	-	-	-	-	-	-	-
QAConv-GS [70]	46.7	75.1	-	-	-	-	-	-	-	-	-	-
MDA † [17]	**53.0**	79.7	-	-	-	-	-	-	-	-	-	-
GN [14]	-	-	-	-	-	-	40.6	67.6	31.2	50.0	11.9	33.4
GN+SNR [14]	37.5	68.0	45.4	66.2	18.3	17.4	-	-	-	-	-	-
LDU [15]	44.8	74.6	48.9	69.2	21.3	21.3	37.5	68.1	29.5	51.8	12.6	36.9
MTI [73]	42.7	72.9	47.7	67.5	16.0	15.4	-	-	-	-	-	-
Baseline (ours)	51.0	76.5	57.1	73.8	32.7	32.9	39.6	66.2	41.4	62.7	16.6	45.2
DCAC (ours)	52.1	77.9	**58.4**	**75.0**	**34.1**	**34.4**	**42.0**	**68.6**	**43.2**	**64.8**	**17.8**	**47.3**

**Table 4 sensors-25-00552-t004:** Comparison with SOTA methods on the multi-source DG Re-ID setting. The best and second-best results are marked with bold and underline, respectively. Data are collected from corresponding works.

Model	Target: MA	Target: D	Target: MS	Target: C3	Average
mAP	R1	mAP	R1	mAP	R1	mAP	R1	mAP	R1
QAConv_50_ [69]	39.5	68.6	43.4	64.9	10.0	29.9	19.2	22.9	28.0	46.6
M^3^L [16]	48.1	74.5	50.5	69.4	12.9	33.0	29.9	30.7	35.4	51.9
M^3^L_*IBN*_ [16]	50.2	75.9	51.1	69.2	14.7	36.9	32.1	33.1	37.0	53.8
RaMoE [50]	56.5	82.0	56.9	73.6	13.5	34.1	35.5	36.6	40.6	56.6
PAT [72]	51.7	75.2	56.5	71.8	21.6	45.6	31.5	31.1	40.3	55.9
DEX [19]	55.2	81.5	55.0	73.7	18.7	43.5	33.8	36.7	40.7	58.9
UDSX [20]	**60.4**	**85.7**	55.8	74.7	20.2	47.6	37.2	38.9	43.4	61.7
SALDG [51]	57.6	82.3	52.0	71.2	18.1	46.5	32.4	34.5	40.0	58.6
DCAC (Ours)	56.7	80.0	**58.9**	**75.4**	**27.5**	**56.7**	**42.5**	**43.6**	**46.4**	**63.9**

**Table 5 sensors-25-00552-t005:** Ablations on various diffusion model fine-tuning methods. θa and θna denote trainable parameters in the attention and non-attention layers of the denoising U-Net. “PT” denotes whether pre-trained diffusion model weights were used. Partial fine-tuning 1 and 2 are the variants of full fine-tuning, where only the output blocks and both the middle and output blocks of the U-Net are trainable, respectively. ✓ and × denote adopting corresponding option or not, respectively. The best results are marked with bold.

Model	PT	Trainable	MA→MS	MS→MA
θa	θna	mAP	R1	mAP	R1
Baseline	-	-	-	22.0	50.1	51.0	76.5
DCAC (frozen)	✓	×	×	21.3	49.7	50.8	76.5
DCAC (LoRA adaptation)	✓	✓	×	**23.4**	**52.1**	**52.1**	77.9
DCAC (partial fine-tuning 1)	✓	✓	✓	22.1	50.3	47.1	74.6
DCAC (partial fine-tuning 2)	✓	✓	✓	22.1	50.4	51.9	**78.5**
DCAC (full fine-tuning)	✓	✓	✓	21.8	50.1	51.5	76.6
DCAC (train from scratch)	×	✓	✓	21.9	50.3	51.7	77.6

**Table 6 sensors-25-00552-t006:** Ablations on the computational overhead of various diffusion model fine-tuning methods. For generalization performance, we use MA→MS results. For computational overhead, we report the time latency and the count of tera floating point operations (TFLOPs) in forward propagation to measure the time efficiency. Additionaly, GPU memory consumption and the number of the model’s trainable parameters are reported to measure space efficiency. The best results are marked with bold.

Mode	Model	MA→MS	Time (ms)	TFLOPs	Memory (GB)	Parameters (M)
mAP	R1
Training	Baseline	22.0	50.1	140.3	1.46	7.58	85.94
DCAC (Frozen)	21.3	49.7	464.7	8.50	17.22	86.51
DCAC (LoRA adaptation)	**23.4**	**52.1**	578.1	8.52	18.15	89.01
DCAC (Partial fine-tuning 1)	22.1	50.3	647.1	8.50	23.92	599.37
DCAC (Partial fine-tuning 2)	22.1	50.4	662.1	8.50	25.01	696.41
DCAC (Full fine-tuning)	21.8	50.1	735.9	8.50	28.24	946.03
Inference	-	-	-	**40.3**	**1.46**	**2.27**	**85.55**

**Table 7 sensors-25-00552-t007:** Ablations on the conditioning scheme. “Instance-wise” denotes that the instance feature is directly adopted as the diffusion condition. “Class-wise” denotes that the classification probability belonging to the real ID class of the image is adopted to generate the diffusion condition. “Correlation-aware” denotes the proposed method, which adopts all ID classification probabilities to generate the diffusion condition in a linear combination manner. The best results are marked with bold.

Conditioning Scheme	MA→MS	MS→MA
mAP	R1	mAP	R1
Baseline	22.0	50.1	51.0	76.5
DCAC (Instance-wise)	22.4	50.8	51.6	77.7
DCAC (Class-wise)	22.6	51.2	50.5	76.5
DCAC (Correlation-aware)	**23.4**	**52.1**	**52.1**	**77.9**

**Table 8 sensors-25-00552-t008:** Ablations on source domain Re-ID performance. The best results are marked with bold.

Model	Market1501	MSMT17
mAP	R1	mAP	R1
Baseline	86.4	94.4	70.4	88.1
DCAC (instance-wise)	86.4	94.7	70.4	**88.5**
DCAC (class-wise)	86.6	94.5	**70.6**	88.2
DCAC (correlation-aware)	**86.8**	**94.9**	70.1	88.3

**Table 9 sensors-25-00552-t009:** Parameter analysis on the rank *r* of LoRA adapters. The best results are marked with bold.

Rank *r*	MA→MS	MS→MA
mAP	Rank-1	mAP	Rank-1
8	**23.4**	**52.1**	51.4	77.3
16	22.1	50.7	51.7	**77.9**
32	22.4	51.4	**52.1**	**77.9**
64	22.9	51.6	51.1	77.5

**Table 10 sensors-25-00552-t010:** Further studies on more sophisticated conditioning schemes beyond the correlation-aware conditioning through linear combination. Non-linear activations like SiLU [77] and normalization layers like batch normalization [78] are adopted on the generated conditions. ✓ and × denote adopting corresponding option or not, respectively. The best results are marked with bold.

Model	Transformations	MA→MS	MS→MA
SiLU	BN	mAP	R1	mAP	R1
Baseline	-	-	22.0	50.1	51.0	76.5
DCAC	×	×	**23.4**	**52.1**	**52.1**	**77.9**
+ Non-linearity	✓	×	22.4	50.8	50.4	76.9
+ BatchNorm	×	✓	22.5	50.9	51.0	76.8
+ ConditionNet	✓	✓	21.9	50.1	50.5	76.2

## Data Availability

All datasets [60,61,62,63] used in this article are publicly accessible.

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
