# Peer review of "Unleashing the Potential of Pre-Trained Diffusion Models for Generalizable Person Re-Identification"

_sensors, 2025, doi:10.3390/s25020552_

Round 1

Reviewer 1 Report

Comments and Suggestions for Authors

1. The manuscript leverages a pre-trained diffusion model as an expert to enhance generalizable feature learning for Domain Generalization Re-Identification(DG Re-ID) and utilizes LoRA adapters for effective fine-tuning. Meanwhile, this paper proposes a correlation-aware conditioning scheme that integrates the dark knowledge embedded in ID classification probabilities with learnable ID-wise prompts to guide the diffusion model. Experiments on both single-source and multi-source DG Re-ID tasks and achieved state-of-the-art performance, and ablation studies verify the effectiveness of the proposed method.

2. However, the paper has the following issues:

(1) Misspelled words: There are some misspelled words in the paper. For example, “multi-modal” in the third paragraph in Introduction should be “multi-model”, “thorough” in Contributions section and “Imagen” in Related Work section. I recommend thoroughly reviewing the manuscript to correct these and any other typographical errors. 

(2) Irregular format: In formula (7), the explanation of the parameter should start with “Where” instead of “Here”.

(3) Explanation of training loss: The section on The Entire Training Loss should be more cohesive. The authors should restate all losses and clarify their composition to ensure smooth and comprehensive understanding.

(4) Table and figure formatting: The scale of Table 2 and Figure 3 is too large, please adjust their size to maintain consistency with the overall manuscript formatting.

(5) Lack of the explanation of LoRA: While the use of LoRA adapters is intriguing, the operating mechanism of LoRA within your model is not sufficiently clear. Although LoRA has shown remarkable results in large language models (LLMs), more detailed explanations specific to its role and function in your model would significantly aid reader comprehension.

Reviewer 2 Report

Comments and Suggestions for Authors

The paper explores the use of pre-trained diffusion models to improve generalizable person re-identification (DG Re-ID), which aims to match individuals across different camera domains without prior exposure to the target domain. It introduces a novel correlation-aware conditioning scheme that integrates classification probabilities and learnable ID-wise prompts to guide the diffusion process, enhancing feature robustness. The proposed method combines a discriminative Re-ID model with the generative capabilities of a diffusion model, achieving state-of-the-art results on single- and multi-source DG Re-ID benchmarks. In general, the paper is well-written, and the novelty is enough. In addition, there are still several weaknesses, as follows:

1. The explanation of the correlation-aware conditioning scheme (Equation 7) is difficult to follow due to ambiguous notation and inadequate context for readers unfamiliar with pre-trained diffusion models. In addition, the integration of LoRA adapters is not sufficiently justified in terms of why this adaptation method was preferred over other fine-tuning techniques.

2. Although the experimental results indicate improvements, some benchmarks (e.g., multi-source DG Re-ID in Table 3) are not explained in sufficient detail. The paper does not discuss the computational overhead introduced by incorporating diffusion models and LoRA adapters.

3.  The introduction provides an adequate background on domain generalizable (DG) person re-identification (Re-ID) but lacks a compelling motivation to demonstrate why the proposed method addresses key limitations in existing works. The novelty claim regarding the use of a diffusion model for DG Re-ID is underexplored. While the paper claims superiority through "correlation-aware conditioning," it lacks a clear comparison to alternative diffusion-based strategies.

4. Some works about Re-ID and detection are suggested to be cited in this paper to make this submission more comprehensive, such as 10.1109/TIP.2024.3514360, 10.1016/j.inffus.2023.102201, 10.1109/TCSVT.2024.3524733, 10.1109/TPAMI.2024.3511621.

Round 2

Reviewer 2 Report

Comments and Suggestions for Authors

No more comments.